# EFFICIENT GRAPH REPRESENTATION LEARNING BY NON-LOCAL INFORMATION EXCHANGE

## ABSTRACT

Graph is an effective data structure to characterize ubiquitous connections as well as evolving behaviors that emerge from the inter-wined system. In spite of various deep learning models for graph data, the common denominator of current state-of-the-art is finding a way to represent, or encode, graph entities (such as nodes and links) on top of the intricate wiring topology. Limited by the stereotype of node-to-node connections, learning global feature representations is often confined in a graph diffusion process where local information has been excessively aggregated as the random walk explores far-reaching neighborhoods on the graph. In this regard, tremendous efforts have been made to alleviate feature oversmoothing issues such that current graph learning backbones can lend themselves in a deep network architecture. However, the focal point of related research often revolves around exploring the expressive power of GNN models, rather than improving the potential of a graph to be expressed, i.e., graph expressibility, which is an intrinsic characteristic of the underlying graph. This is not only more relevant for the downstream applications but also more effective to mitigate the oversmoothing risk by changing the expressibility to reduce unnecessary information exchange on the graph. Inspired by the notion of non-local mean techniques in image processing area, we propose a non-local information exchange mechanism by establishing an express connection to the distant nodes, instead of propagating information along the (possibly very long) topological pathway node-after-node. Since the seek of express connections throughout the graph could be computationally expensive in real-world applications, we further present a hierarchical re-wiring framework (coined *express messenger* wrapper) to progressively incorporate express links into graph learning in a local-to-global manner, which allows us to effectively capture multi-scale graph feature representations without using a very deep model, thus free of the over-smoothing challenge. We have integrated our *express messenger* wrapper (as a model-agnostic plug-in) with existing graph neural networks (either using graph convolution or transformer backbones) and achieved SOTA performance on various graph learning applications.

## 1 INTRODUCTION

Graph is a universal data structure to model complex relations in the real world, from social network connections to protein interactions. In the past decade, remarkable strides were made in the development of graph neural networks (GNNs) for node classification Kipf & Welling (2016), link prediction Zhang & Chen (2018), and graph recognition Xu et al. (2019). Along with the trend in computer vision, the research focus has shifted to transformer-based backbone following the widespread adoption of graph convolution techniques Yun et al. (2019); Kreuzer et al. (2021), where global attention mechanism Vaswani et al. (2017) demonstrate great potential to alleviate longstanding issues in GNN such as over-smoothing and poor long-range feature dependencies Chen et al. (2020). To address the computational challenge of large-scale graphs, tokenized graph transformer Kim et al. (2022) models have been introduced recently where the graph token is localized either at the node level Chen et al. (2022) or within a sub-graph He et al. (2023b).

Despite various deep models for graph learning, current GNNs are *de facto* closely bonded under the overarching umbrella of a topological message-passing paradigm Cai et al. (2021); Feng et al. (2022). Given the observed graph feature representations, the driving factor of GNN is to learn

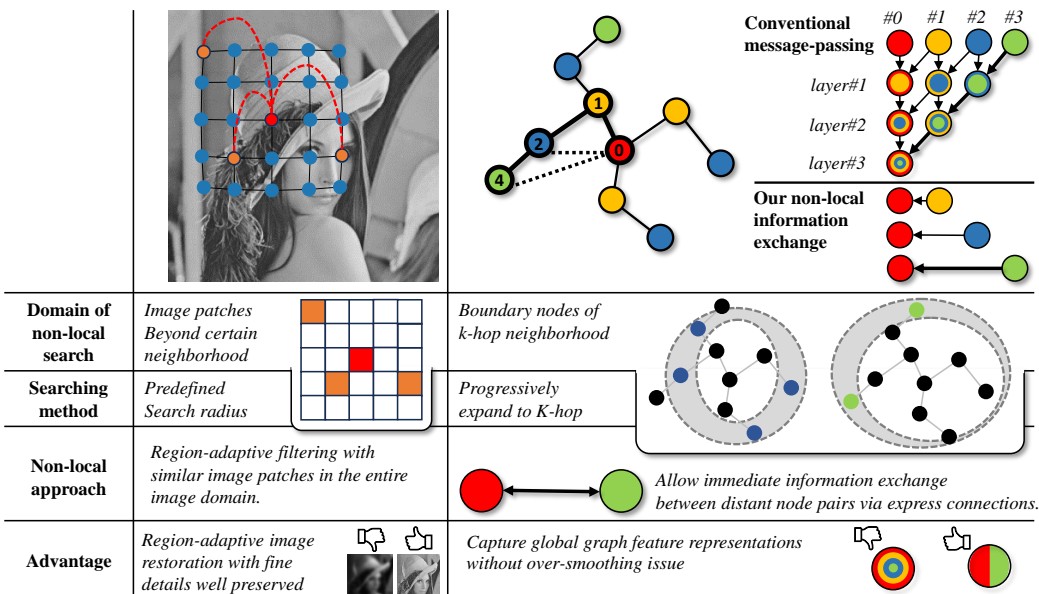

Figure 1: Non-local information exchange mechanism (right), a reminiscent of non-local mean technique for image processing (left), is able to capture global information by express connections (dashed links), which reduces the over-smoothing risk in GNN. Both ideas integrate information beyond either a spatial or topological neighbor, in order to preserve distinctive feature representations.

intrinsic feature representations by alternatively (1) seeking for an optimal feature subspace and (2) aggregating the information within a certain graph neighborhood, where the learned feature representations are supposed to have a better alignment with the existing labels (supervised manner Veličković et al. (2017)) or exhibit a more structured behavior (unsupervised manner Mo et al. (2022)). Since the node-to-node information exchange fundamentally underlines the graph topology, the wiring pattern presented in the graph becomes a pivotal factor steering the expressive power (explained in Section 2.2) of GNN models and thus influencing the graph learning performance Li & Leskovec (2022); Morris et al. (2019); Balcilar et al. (2021).

There is a converging consensus that the message-passing mechanism allows us to capture global graph feature representations, in a layer-by-layer fashion, by progressively aggregating the feature representations from the nearby nodes to the distant nodes. Limited by the pairwise links in the graph[1], however, the cost of such node-after-node message passing is over-smoothed feature representations due to an excessive number of feature aggregations along the pathway between two distant nodes on the graph Alon & Yahav (2020); Topping et al. (2021). As demonstrated in the top-right panel of Figure 1, local features (centered at node #0 and #1) have to be aggregated multiple times until the message-passing process reaches node #3. In this context, the combination of local and global information by conventional node-after-node message-passing mechanism often yields over-smoothed feature representations which mixed with all information spanning the entire graph pathway.

Inspired by the non-local means technique in patch-based image processing Buades et al. (2005), we introduce a non-local exchanging (NLE) mechanism to the field of GNN by establishing express connections for distant node pairs on top of the conventional message-passing *cliché* in GNN. In Figure 1 bottom-right, the selected express connection not only allows us to effectively capture global information but also offers a new window to maintain the distinctive power of the underlying feature representations. It is worth noting that our idea of express connection is simple yet effective. The conjecture is that the graph re-wiring step enhances the expressibility of graph topology, thus enabling GNN to learn informative global features prior to the incoming features undergoing significant smoothing by conventional message-passing routine. To that end, deep network architecture is no longer the only option to capture global feature representations, thus reducing the over-smoothing risk in GNN. In this context, we further devise a hierarchical re-wiring plug-in, called *express messenger* (*ExM*) wrapper, that naturally fits the layer-by-layer network architecture of GNN models. The SOTA performance on various graph applications demonstrates that the novel NLE mechanism

---

[1]High-order network models such as hyper-graph technique use hyperlink to model relationship among multiple nodes. However, most graph applications are conducted on 1-simplex graphs Battiston et al. (2020).

unleashes the power of current GNN models including graph convolution networks (GCN) and graph transformer models.

Our main contributions are summarized below:

- We propose a non-local information exchange mechanism to efficiently integrate feature representations from non-local neighborhoods by a collection of express connections.

- Current works focus on the optimization of feature representation learning by overcoming the over-smoothing issue. We address this challenge in a novel perspective of expressibility of graph topology, i.e., the insight of our NLE mechanism is to directly combine local and global information through express links before the over-smoothed features undermine the discriminative power of feature representation.

- We devise our *ExM* wrapper as an agnostic GNN plug-in, which facilitates various GNN models to retain SOTA performance on both homophilous and heterophilous graph data.

## 2 PRELIMINARIES

Let graph $\mathcal{G} = \{\mathcal{V}, \mathcal{E}\}$, where $\mathcal{V}$ is node set and $\mathcal{E}$ is edge set. The strength of node-to-node connectivity is encoded in a weighted adjacency matrix $\mathbf{A}$. Specifically, we use $\mathcal{N}_i^k$ to denote the graph neighborhood of nodes where the topological distance with respect to the $i$-th node is $k$-hops on the graph $\mathcal{G}$. Note, $\mathcal{N}_i^0$ indicates direct-connected nodes from the $i$-th node without any hops. Following the notion in GNN literature, the latent graph feature representations at the $l$-th layer of neural network is denoted by $\mathbf{H}^{(l)}$.

### 2.1 RELATED WORKS

In general, graph feature representation can be formulated by $\mathbf{H}^{(l+1)} = \mathcal{F}_{\mathbf{W}}(\mathbf{A}, \mathbf{H}^{(l)})$, where $\mathcal{F}$ is a model-specific learnable function parameterized by $\mathbf{W}$. Taking vanilla GCN Kipf & Welling (2016) as an example, $\mathcal{F}_{\mathbf{W}} = \mathbf{A}(\mathbf{HW})$ alternatively (1) projects the current feature representations to a new subspace by $\mathbf{HW}$ and (2) aggregates the updated features with the graph neighborhoods $\{\mathcal{N}_i^0\}_{i=1}^{|\mathcal{V}|}$ underlying the graph topology steered by $\mathbf{A}$. It is apparent that node features eventually become identical after enough number of graph diffusion steps Coifman & Lafon (2006), which is the smoking gun of over-squashing and over-smoothing in GNN models Alon & Yahav (2020); Topping et al. (2021).

In this regard, striking efforts have been made to preserve the local structures. Since all neural networks are driven by gradients, the gradient gating technique Rusch et al. (2022) is proposed to alleviate over-smoothed graph features at each node by penalizing message passing between two nodes bearing distinct features. Alternatively, there have been several interesting works addressing the same problem by re-wiring the graph topology. For instance, DropEdge Rong et al. (2019) randomly removes a certain number of edges from the input graph at each training epoch, in order to avoid the excessive message passing. On the contrary, a fully-adjacent (FA) layer is introduced in Alon & Yahav (2020) where every pair of nodes is connected by an (immediate) edge. Although the FA layer is empirically effective in preventing over-smoothing by easing the bottleneck of information flow, connecting all pairs of nodes without any selection significantly contradicts the inductive bias inherent in graph data learning Battaglia et al. (2018). In between these two extreme ends, the sparse technique is used in graph diffusion convolution (GDC) Gasteiger et al. (2019) to obtain more localized graph convolution. More advanced edge-based combinatorial curvature is introduced Topping et al. (2021) which formulates message passing in graph learning as a stochastic discrete Ricci flow (SDRF). As summarized in the bottom of Figure 1, our proposed non-local information exchange mechanism inherits the benefits of the aforementioned topology re-wiring techniques while also being more general and agnostic to improve the current GNN models instantly as a seamless plug-in component.

Alongside the streamlined evolution of transformers in computer vision Dosovitskiy et al. (2010); Liu et al. (2021), research focus has shifted to enhancing long-range feature representations in graph-based transformers by capitalizing on the self-attention mechanism in the transformer backbone Vaswani et al. (2017). Specifically, the instance of graph token can be defined at multiple levels

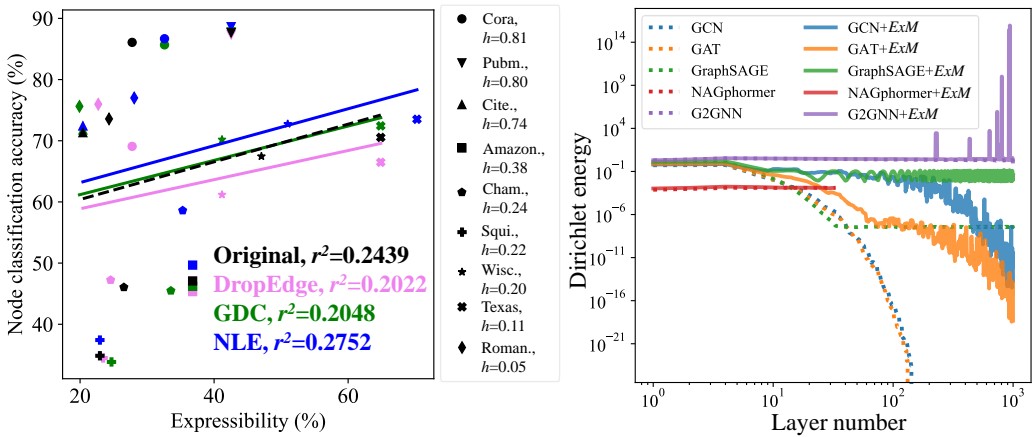

Figure 2: Left: Relationship between graph expressibility and GNN performance (in node classification accuracy). We use landmark patterns to represent different graph data sets and colors for topology re-wiring methods, respectively. Note that all rewiring methods are applied with the same hyperparameter of baseline. Right: Over-smoothness is delayed after graph rewiring by our *ExM*.

(node, edge, or sub-graph Kim et al. (2022); He et al. (2023a); Chen et al. (2022); Zhang et al. (2022; 2020)), which allows us to characterize the interactions between multi-scale graph feature representations across a wide range of neighborhood sizes. Even though local and global features exhibit distinct characteristics, current graph transformer models tend to mix these multi-scale features within the self-attention module. Notably, while the topology re-wiring technique has proven effective in vanilla GNN models, it remains unexplored in the context of graph transformer models.

## 2.2 PROXY MEASUREMENT FOR GRAPH EXPRESSIBILITY

1-dimension Weisfeiler-Lehman (WL) algorithm, aka. node-wise color refinement algorithm, is a classic node classification approach used for testing whether two graphs are isomorphic or not Morris et al. (2019). In Morris et al. (2021), it is observed that color-coded labels through the high-order WL algorithm (coloring graph tuple by tuple) often yield a higher graph prediction accuracy. Following the notion in Morris et al. (2021), we define a proxy measure of graph expressibility by (1) calculating node features based on the hashing code of coloring labels by 1-WL algorithm, (2) training an SVM to predict the node label, and (3) use the prediction accuracy as the degree of graph expressibility. Since the nature of WL algorithm underlines the graph topology, the graph expressibility proxy is a good indicator of how graph topology affects learning performance.

As a proof-of-concept experiment, we fit the relationship between graph expressibility proxy on nine popular graph datasets (Roman, Texas, Wisconsin, Squirrel, Chameleon, Amazon, CiteSeer, PubMed, and Cora) and node prediction accuracy by NAGphomer Chen et al. (2022) (a latest graph transformer model). Specifically, we sort these nine graph datasets based on a homophily metric $h$ ratio (Figure 2 right) by Zhu et al. (2020), where higher degree suggests the nodal label and underlying graph topology are more consistent. In general, the linear regression results (black landmarks and black dash line in Figure 2) indicate that the higher the graph expressibility, the greater the potential for improved graph learning performance.

## 2.3 TOPOLOGY RE-WIRING VS. GRAPH EXPRESSIBILITY

Furthermore, we calculate the graph expressibility proxy for the re-wired graphs after applying three existing re-wiring methods DropEdge and GDC (reviewed in Section 2.1). Due to space limit, we only display the graph expressibility proxies for the five least homophilic graphs (i.e., bottom five smallest $h$) in Table 1. Surprisingly, (nearly) none of the current re-wiring methods can enhance the graph expressibility. Based on the observed relationship between graph expressibility and GNN performance, the reduced expressibility degree seems to indicate worse graph learning performance after applying current re-wiring methods. Regrettably, this supposition has been confirmed by the pink (by DropEdge) and green (by GDC) lines in Figure 2 left part: The node prediction accuracy on re-wired graph topology is consistently lower (at least equal) than training the same GNN model on the original graph, implying that the altered graph topology by current re-wiring methods actually undermines the graph learning performance.

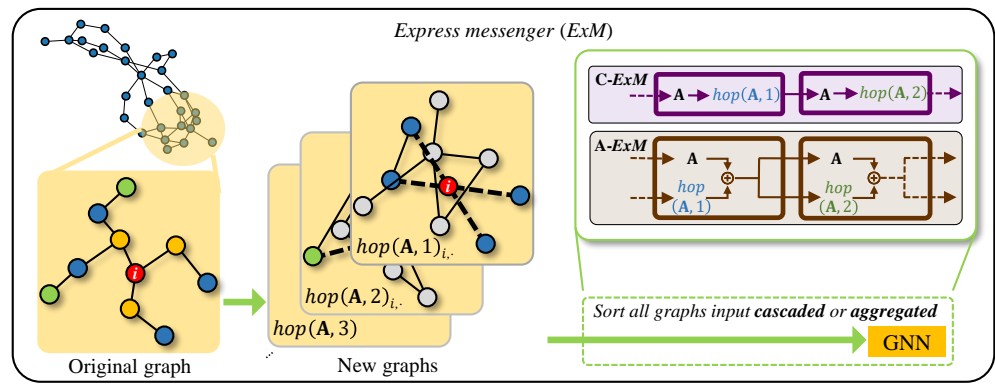

Figure 3: Left: Illustration of NLE for a simulated graph from original with adjacency matrix $\mathbf{A}$ to re-wired with a new adjacency $hop(\mathbf{A}, k)$ ($k = 1, 2, 3$). Right: *ExM* sorts the original graph and new graphs cascaded (C-*ExM*) or aggregated (A-*ExM*) to input to any GNN.

In the right part of Figure 2, we follow the experiment by Rusch et al. (2022) to explore the effection on over-smoothness. Given that solid curves are all fallen slower than dashed, our *ExM* changes the graph topology to far-away nodes can mitigate the over-smoothing issue without a specific GNN design.

## 3  METHODS

To unleash the expressibility of a given graph topology, we present a novel non-local information exchange mechanism which is agnostic to various GNN models. As explained in Figure 1, our core idea is to capitalize on a collection of express connections for capturing multi-scale graph feature representations, thus being free of over-smoothed features due to the conventional diffusion-based message-passing mechanism.

### 3.1  GRAPH REWIRING BY HIERARCHICAL NON-LOCAL INFORMATION EXCHANGE

**Exhaustive vs. progressive NLE.** Supposing the non-local search is performed throughout the entire graph, the exhaustive NLE mechanism yields a complementary graph topology $\mathbf{1} - \mathbf{A}$ in addition to original graph $\mathbf{A}$, where $\mathbf{1}$ denotes the all-one matrix. Since most intelligent systems process information in a hierarchical manner Goodfellow et al. (2016), we present a progressive NLE that gradually increases the hopping steps $k$ and includes distant (but topologically connected) nodes $v_j$ for the underlying node $v_i$ ($i \neq j$) by a node selection function (1: selected for express connection; 0: otherwise):

$$hop(\mathbf{A}, k)_{ij} = \begin{cases} 1 & \text{for} \quad j \in \mathcal{N}_i^k - \mathcal{N}_i^{k-1} \\ 0 & \text{for} \quad j \notin \mathcal{N}_i^k - \mathcal{N}_i^{k-1} \end{cases} \tag{1}$$

where the hopping steps $k$ acts as the the search radius from $v_i$ to $v_j$ on the graph.

In Figure 3, we use a simulated graph (left-top) to demonstrate the re-wiring results by progressive NLE, the Express Messenger (*ExM*). Specifically, the re-wired graph topologies by applying $hop(\mathbf{A}, 1)$ and $hop(\mathbf{A}, 2)$ to the original adjacency matrix $\mathbf{A}$ are shown at Figure 3. To clearly display the express connections at each graph node, we zoom into a sub-graph and show the established express connections at $i$-th node (dash lines of new graphs). The benefit of our progressive NLE has

Table 1: Comparing graph expressibility changes by related re-wiring methods and our NLE.

|  | $h$ | Original | DropEdge | GDC | NLE (Ours) |
|---|---|---|---|---|---|
| Chameleon | 0.24 | 26.54 | 24.56 (-1.98) | 33.55 (+7.01) | 35.31 (+8.77) |
| Squirrel | 0.22 | 22.96 | 23.44 (+0.48) | 22.96 (-0.00) | 22.96 (-0.00) |
| Wisconsin | 0.20 | 47.06 | 41.18 (-5.88) | 41.18 (-5.88) | 50.98 (+3.92) |
| Texas | 0.11 | 64.87 | 64.87 (-0.00) | 64.87 (-0.00) | 70.27 (+5.40) |
| Roman-empire | 0.05 | 24.32 | 22.71 (-1.61) | 19.91 (-4.41) | 28.08 (+3.76) |

been also validated on the real-world data, as the increased graph expressibility values shown in the last column of Table 1. As anticipated, we have observed significantly enhanced node prediction accuracy by applying GNN model on our re-wired graph (blue landmarks and line in Figure 2).

It is worth noting that, by increasing the hopping step $k$ to the longest graph path $K$, the total union of $hop(\mathbf{A}, k)$ is equivalent to exhaustive NLE.

**Proposition 3.1.** *The formula 1 converges on $\sum_{k=1}^{K} hop(\mathbf{A}, k) = \mathbf{1} - \mathbf{A}$.*

The proof is derived as:

$$\mathbf{1} - \mathbf{A} = \mathcal{N}^K - \mathcal{N}^{K-1} + \mathcal{N}^{K-1} - \mathcal{N}^{k-2} + \cdots + \mathcal{N}^1 - \mathcal{N}^0$$

$$= hop(\mathbf{A}, K) + hop(\mathbf{A}, K-1) + \cdots + hop(\mathbf{A}, 1) = \sum_{k=1}^{K} hop(\mathbf{A}, k).$$

### 3.2 EXPRESS MESSENGER WRAPPER: AN AGNOSTIC GNN PLUG-IN

It is evident that the original adjacency matrix $\mathbf{A}$ (node-to-node topology used in conventional GNN models) and rewired topology $\bigcup_{k=1}^{K} hop(\mathbf{A}, k)$ (non-local topology re-wired by NLE) represent completely different aspects of graph topology. In this context, we devise two topology-wrapping network architectures that allow us to tailor our NLE mechanism to a variety of current GNN models.

In both wrapping methods, we wrap two GNN backbones (MLP layers in vanilla GNN and tokens in graph transformer model) into one building block: One uses original graph $\mathbf{A}$ and the other uses the $k$-specific topology of express connections only. The major difference lies in the information fusion mechanism, as described below.

**Cascaded express messenger (C-*ExM*) wrapper.** By fixing the hopping step $k$, we first train a GNN backbone to learn feature representations using the original graph topology $\mathbf{A}$, i.e., $\mathbf{H}^{(l+1)} \leftarrow \mathcal{F}_{\mathbf{W}}(\mathbf{A}, \mathbf{H}^{(l)})$. Next, the updated $\mathbf{H}^{(l+1)}$ becomes the input to a cascaded GNN backbone to learn global feature representations delivered by the express connections, i.e., $\mathbf{H}^{(l+2)} \leftarrow \mathcal{F}_{\mathbf{W}}(hop(\mathbf{A}, k), \mathbf{H}^{(l+1)})$. The step-by-step implementation is summarized in Algorithm 1, along with the graphic diagram shown in the middle of Figure 3. Note, in case the predefined layer number $L$ is greater than the largest hopping step, our C-*ExM* wrapper skips GNN backbone using re-wired topology, thus becoming the conventional GNN model.

**Aggregated express messenger (A-*ExM*) wrapper.** Instead of alternatively updating feature representations in C-*ExM* wrapper, we train two identical GNN backbones using original topology $\mathbf{A}$ and re-wired topology $hop(\mathbf{A}, k)$ in parallel. As the graphic diagram shown in the middle of Figure 3, we combine the output from two GNN backbones by averaging (details in Algorithm 2).

---

| **Algorithm 1** Cascaded *ExM* wrapper | **Algorithm 2** Aggregated *ExM* wrapper |
|---|---|
| **Require:** Range of hopping steps $[K_{start}, K_{end}]$
$\quad$ hop$(\mathbf{A}, k) \leftarrow Eq.\ 1, \forall k \in [K_{start}, K_{end}]$
$\quad$ **while** $l = 1 : L$ **do**
$\quad\quad$ **if** $l$ is odd **or** $l/2 > (K_{end} - K_{start})$ **then**
$\quad\quad\quad \mathbf{H}^{(l+1)} \leftarrow \mathcal{F}_{\mathbf{W}}(\mathbf{A}, \mathbf{H}^{(l)})$
$\quad\quad$ **else if** $l$ is even **then**
$\quad\quad\quad \mathbf{H}^{(l+1)} \leftarrow \mathcal{F}_{\mathbf{W}}(hop(\mathbf{A}, l/2), \mathbf{H}^{(l)})$
$\quad\quad$ **end if**
$\quad$ **end while** | **Require:** Range of hopping steps $[K_{start}, K_{end}]$
$\quad$ hop$(\mathbf{A}, k) \leftarrow Eq.\ 1, \forall k \in [K_{start}, K_{end}]$
$\quad$ **while** $l = 1 : L$ **do**
$\quad\quad \mathbf{H}^{(l+1)} \leftarrow \mathcal{F}_{\mathbf{W}}(\mathbf{A}, \mathbf{H}^{(l)})$
$\quad\quad$ **while** $k = K_{start} : K_{end}$ **do**
$\quad\quad\quad \mathbf{H}^{(l+1)} \leftarrow \mathbf{H}^{(l+1)} \oplus \mathcal{F}_{\mathbf{W}}(hop(\mathbf{A}, k), \mathbf{H}^{(l)})$
$\quad\quad$ **end while**
$\quad$ **end while** |

---

## 4 EXPERIMENTS

We evaluate our proposed *ExM* wrapper through two types of experiments: (1) Benchmark graph rewiring performance by comparing to prior rewiring methods. (2) Benchmark graph representation learning performance on node classification using our *ExM* wrapper. We conduct some ablation studies to further assess the effectiveness of our *ExM* wrapper.

### 4.1 EXPERIMENTAL SETTING

**Dataset.** Experiments are carried out on a set of nine publicly accessible graph datasets, encompassing three homophilous graphs (Cora, PubMed, and Citeseer) and six heterophilous graphs (Texas, Wisconsin, Chameleon, Squirrel, Roman-empire (Roman.), and Amazon-ratings (Amazon.)). The initial seven graphs are widely recognized for evaluating graph representation learning techniques, while the last two were introduced by Platonov et al. (2023) and are distinguished by their nodes without duplication. For the evaluation of graph representation learning, we employ 10 random splits (train:val:test = 6:2:2) for each of the first seven graphs, as suggested by Pei et al. (2020). Among the five new graph datasets introduced by Platonov et al. (2023), we utilize two of them. The remaining three datasets, which feature only two classes, are specifically employed for our ablation studies. Our split settings are consistent with those outlined by Platonov et al. (2023). For further details regarding the profiles of the graph datasets employed, please refer to Appendix A.

**Experiment setup.** We conduct experiments using six baseline graph neural networks to evaluate our *ExM* framework: graph attention networks (GAT) Veličković et al. (2017), graph transformer networks (GT) Shi et al. (2020), GraphSAGE (referred to as SAGE) Hamilton et al. (2017), NAG-phormer (referred to as NAG) Chen et al. (2022), JacobiConv Wang & Zhang (2022), and feature selection graph neural network (FSGNN) Maurya et al. (2022). Among these, GAT, GT, and SAGE implementations are provided by Platonov et al. (2023), while the rest are implemented by their respective studies. We set the hyperparameters through the best model they provided.

Our *ExM* wrapper is employed without altering the model architecture to ensure a fair comparison. This implies that we employ the same number of GNN layers, even with the inclusion of the *ExM* wrapper. The performance is demonstrated using the optimal combination of hyperparameters: $K_{start}$, $K_{end}$, and exhaustive or progressive NLE mechanism. The impact of varying these hyperparameters is detailed in our ablation studies. Additionally, we present results along with the graph homophily ratio $h$, in line with the approach outlined in Zhu et al. (2020), providing further insights on how the characteristics of graph topology influence graph learning.

### 4.2 RESULTS

#### 4.2.1 BENCHMARK GRAPH RE-WIRING TECHNIQUES

Since transformer models start to prevail in GNN field, we select the latest graph transformer model NAG as the reference to benchmark the effect of various graph re-wiring techniques, which include DropEdge, GDC, SDRF, C-*ExM$_E$* (cascaded with exhaustive NLE), A-*ExM$_E$* (aggregated with exhaustive NLE), C-*ExM$_P$* (cascaded with progressive NLE), and A-*ExM$_P$* (aggregated with progressive NLE).

Table 2: Benchmark results for graph rewiring performance. NAG is the baseline here. Red denotes the first rank, followed by blue ($2^{nd}$), and violet ($3^{rd}$). "E" means exhaustive NLE, and "P" is progressive NLE. '–' means the model has no result reported.

| | Roman. $h$=0.05 | Texas $h$=0.11 | Wisconsin $h$=0.20 | Squirrel $h$=0.22 | Chameleon $h$=0.24 | Amazon. $h$=0.38 | Citeseer $h$=0.74 | Pubmed $h$=0.80 | Cora $h$=0.81 |
|---|---|---|---|---|---|---|---|---|---|
| Baseline | $73.57_{\pm1.30}$ | $70.54_{\pm3.07}$ | $67.45_{\pm1.80}$ | $34.85_{\pm0.85}$ | $46.05_{\pm1.10}$ | $47.08_{\pm0.60}$ | $71.32_{\pm0.65}$ | $87.65_{\pm0.24}$ | $86.08_{\pm0.69}$ |
| DropEdge | $75.97_{\pm0.83}$ | $66.48_{\pm3.24}$ | $61.18_{\pm2.12}$ | $34.49_{\pm1.13}$ | $47.23_{\pm1.09}$ | $45.34_{\pm0.50}$ | $72.40_{\pm0.46}$ | $87.33_{\pm0.46}$ | $69.08_{\pm0.67}$ |
| GDC | $75.62_{\pm0.26}$ | $72.43_{\pm1.62}$ | $70.19_{\pm3.37}$ | $33.84_{\pm0.93}$ | $45.48_{\pm1.09}$ | $46.20_{\pm0.55}$ | $71.76_{\pm0.58}$ | $87.80_{\pm0.28}$ | $85.67_{\pm0.75}$ |
| SDRF[2] | – | $70.35_{\pm0.60}$ | $61.55_{\pm0.86}$ | $37.67_{\pm0.23}$ | $44.46_{\pm0.17}$ | – | $72.58_{\pm0.20}$ | $79.10_{\pm0.11}$ | $82.76_{\pm0.23}$ |
| C-*ExM$_E$* | $73.99_{\pm0.49}$ | $73.51^*_{\pm2.02}$ | $72.75^*_{\pm2.05}$ | $35.47_{\pm1.25}$ | $48.86_{\pm1.12}$ | $46.79_{\pm0.70}$ | $71.59_{\pm0.64}$ | $88.64_{\pm0.38}$ | $86.40_{\pm0.37}$ |
| A-*ExM$_E$* | $76.51^*_{\pm0.72}$ | $71.62_{\pm3.02}$ | $70.79_{\pm0.50}$ | $34.98_{\pm0.69}$ | $47.39_{\pm1.60}$ | $47.34_{\pm0.67}$ | $72.43_{\pm0.76}$ | $87.96_{\pm0.24}$ | $86.68_{\pm0.55}$ |
| C-*ExM$_P$* | $69.78_{\pm0.99}$ | $72.70_{\pm3.07}$ | $62.55_{\pm0.43}$ | $37.42^*_{\pm1.19}$ | $48.93_{\pm1.53}$ | $49.67_{\pm0.48}$ | $70.92_{\pm0.39}$ | $87.87_{\pm0.32}$ | $86.58_{\pm0.25}$ |
| A-*ExM$_P$* | $77.00^*_{\pm0.59}$ | $68.92_{\pm3.25}$ | $54.90_{\pm0.88}$ | $34.49_{\pm1.01}$ | $58.60^*_{\pm1.12}$ | $48.33_{\pm0.46}$ | $71.70_{\pm0.76}$ | $87.18_{\pm0.40}$ | $86.42_{\pm0.47}$ |

Table 2 presents the results of graph rewiring performance across nine node classification tasks. Leveraging non-local information has led to improvements in node classification across various graph types. For instance, on homophilous graphs such as Cora ($h$=0.81) and Pubmed ($h$=0.80), as well as on heterophilous graphs like Roman-empire ($h$=0.05), the baseline performance has been

---

[2]SDRF builds upon the foundation of GNN, and the reported results use the same splitting ratios as ours.

improved. Among five heterophilous graphs ($h \leq 0.24$), four are improved significantly (denoted by '*'). With the exception of Squirrel and Citeseer, where our method achieved the second-best performance, it secured the top-3 position in all other cases. In contrast, previous graph re-wiring approaches faced challenges with homophilous graphs and exhibited lower accuracy on graphs with $h > 0.24$.

### 4.2.2 EVALUATION ON GRAPH FEATURE REPRESENTATION LEARNING

Table 3 illustrates the contrast between the baseline GNN model and baseline+*ExM* wrapper, where our *ExM* wrapper facilitates most baseline methods retaining SOTA performance. For clarity, we only report the highest performance among four variations of *ExM* wrapper. However, it's worth noting that in the majority of cases, the C-*ExM*$_P$ wrapper outperforms the other three counterparts. Specifically, the *ExM* wrapper allows us to successfully secure a top-3 ranking across seven different graph datasets characterized by diverse homophily ratios $h$. The *ExM* wrapper is seamlessly integrated into various baselines, ranging from GAT to the recent FSGNN.

In addition, we include results from four very recent SOTA methods: GPRGNN Chien et al. (2020), UGT Hoang et al. (2023), GloGNN Li et al. (2022), and GBKGNN Du et al. (2022), alongside the conventional GCN with 5 layers for comprehensive comparison. Consistent with the previous experiment, GNN models with the *ExM* wrapper often demonstrate more pronounced improvements in performance for graphs, especially with smaller $h$ values. This aligns with the established understanding that heterophilous graphs necessitate specific model architectures for achieving SOTA performance. Conversely, we showcase that leveraging non-local information exchange leads to even higher accuracy, underscoring the efficiency of this approach in graph representation learning. Meanwhile, it's worth noting that SAGE+*ExM* achieves a new performance record on the Roman-empire dataset, as reported in the current leaderboard by Platonov et al. (2023).

Table 3: Performance by using NLE to other SOTA methods. Red denotes the first rank, blue the second, and violet the third. Upper results are exacted from the literature, bottom is implemented by us. '−' means the model has no result reported.

| | Roman. $h=0.05$ | Texas $h=0.11$ | Wisconsin $h=0.20$ | Squirrel $h=0.22$ | Chameleon $h=0.24$ | Citeseer $h=0.74$ | Pubmed $h=0.80$ |
|---|---|---|---|---|---|---|---|
| GAT | $80.92_{\pm0.68}$ | $58.92_{\pm5.81}$ | $60.39_{\pm3.67}$ | $62.00_{\pm1.29}$ | $69.85_{\pm1.72}$ | $74.03_{\pm1.23}$ | $87.86_{\pm0.42}$ |
| GAT+*ExM* | $86.06_{\pm0.35}$ | $75.14_{\pm7.73}$ | $72.35_{\pm7.36}$ | $69.54_{\pm1.30}$ | $73.18_{\pm1.60}$ | $74.25_{\pm1.27}$ | $88.04_{\pm0.42}$ |
| GT | $85.70_{\pm0.99}$ | $62.43_{\pm7.80}$ | $57.65_{\pm4.91}$ | $58.09_{\pm1.50}$ | $68.99_{\pm2.55}$ | $74.68_{\pm1.46}$ | $87.51_{\pm0.52}$ |
| GT+*ExM* | $89.20_{\pm0.63}$ | $68.11_{\pm10.1}$ | $67.06_{\pm8.71}$ | $58.09_{\pm1.50}$ | $70.92_{\pm1.44}$ | $74.83_{\pm1.22}$ | $87.51_{\pm0.52}$ |
| SAGE | $86.96_{\pm0.56}$ | $80.27_{\pm5.70}$ | $81.37_{\pm5.49}$ | $44.53_{\pm1.08}$ | $62.92_{\pm1.70}$ | $75.19_{\pm1.51}$ | $88.59_{\pm0.38}$ |
| SAGE+*ExM* | $90.34_{\pm0.42}$ | $82.97_{\pm6.38}$ | $84.71_{\pm2.23}$ | $45.49_{\pm1.17}$ | $62.92_{\pm1.70}$ | $75.51_{\pm1.46}$ | $88.75_{\pm0.38}$ |
| NAG | $73.57_{\pm1.30}$ | $70.54_{\pm3.07}$ | $67.45_{\pm1.80}$ | $34.85_{\pm0.85}$ | $46.05_{\pm1.10}$ | $71.32_{\pm0.65}$ | $87.65_{\pm0.24}$ |
| NAG+*ExM* | $77.00_{\pm0.59}$ | $73.51_{\pm2.02}$ | $72.75_{\pm2.05}$ | $37.42_{\pm1.19}$ | $58.60_{\pm1.12}$ | $72.43_{\pm0.68}$ | $88.64_{\pm0.38}$ |
| JacobiConv | $71.25_{\pm0.45}$ | $76.49_{\pm6.74}$ | $76.67_{\pm5.00}$ | $50.21_{\pm2.39}$ | $67.94_{\pm1.13}$ | $76.00_{\pm1.44}$ | $88.95_{\pm0.46}$ |
| JacobiConv+*ExM* | $72.75_{\pm0.64}$ | $79.46_{\pm3.86}$ | $79.22_{\pm4.49}$ | $53.07_{\pm2.54}$ | $69.89_{\pm1.50}$ | $76.08_{\pm1.54}$ | $89.14_{\pm0.42}$ |
| FSGNN | $67.93_{\pm0.53}$ | $85.95_{\pm5.77}$ | $85.88_{\pm4.94}$ | $73.55_{\pm2.16}$ | $78.16_{\pm1.11}$ | $76.52_{\pm1.76}$ | $89.63_{\pm0.40}$ |
| FSGNN+*ExM* | $72.61_{\pm0.57}$ | $86.49_{\pm4.52}$ | $87.06_{\pm3.84}$ | $73.85_{\pm2.08}$ | $78.22_{\pm0.81}$ | $76.95_{\pm1.25}$ | $89.72_{\pm0.49}$ |
| GCN | $73.69_{\pm0.74}$ | $59.46_{\pm5.25}$ | $59.80_{\pm6.99}$ | $36.89_{\pm1.34}$ | $59.82_{\pm2.58}$ | $76.68_{\pm1.64}$ | $87.38_{\pm0.66}$ |
| GPRGNN | $64.85_{\pm0.27}$ | $92.92_{\pm0.61}$ | − | $49.93_{\pm0.53}$ | $67.48_{\pm0.40}$ | $67.63_{\pm0.38}$ | $85.07_{\pm0.09}$ |
| UGT | − | $86.67_{\pm8.31}$ | $81.60_{\pm8.24}$ | $66.96_{\pm2.49}$ | $69.78_{\pm3.21}$ | $76.08_{\pm2.50}$ | − |
| GloGNN | $59.63_{\pm0.69}$ | $84.32_{\pm4.15}$ | $87.06_{\pm3.53}$ | $57.54_{\pm1.39}$ | $69.78_{\pm2.42}$ | $77.41_{\pm1.65}$ | $89.62_{\pm0.35}$ |
| GBKGNN | $74.57_{\pm0.47}$ | $81.08_{\pm4.88}$ | $84.21_{\pm4.33}$ | − | − | $79.18_{\pm0.96}$ | $89.11_{\pm0.23}$ |

### 4.2.3 REPRESENTATION VISUALIZATION

In Figure 4, we apply t-distributed Stochastic Neighbor Embedding (t-SNE) Van der Maaten & Hinton (2008) to project the high-dimensional features into a 2D space, allowing for visual inspection of the representation changes induced by *ExM* wrapper. Each colored dot represents an individual node. Using SAGE as an illustrative example with $K_{start} = 1$ and $K_{end} = 2$, we can observe notable shifts in the distribution of orange dots in the $2^{nd}$ and $4^{th}$ layers after topological re-wiring by *ExM*. In contrast, the baseline model struggles to stratify these nodes, and its feature distribution lacks the pronounced diversity seen in our approach.

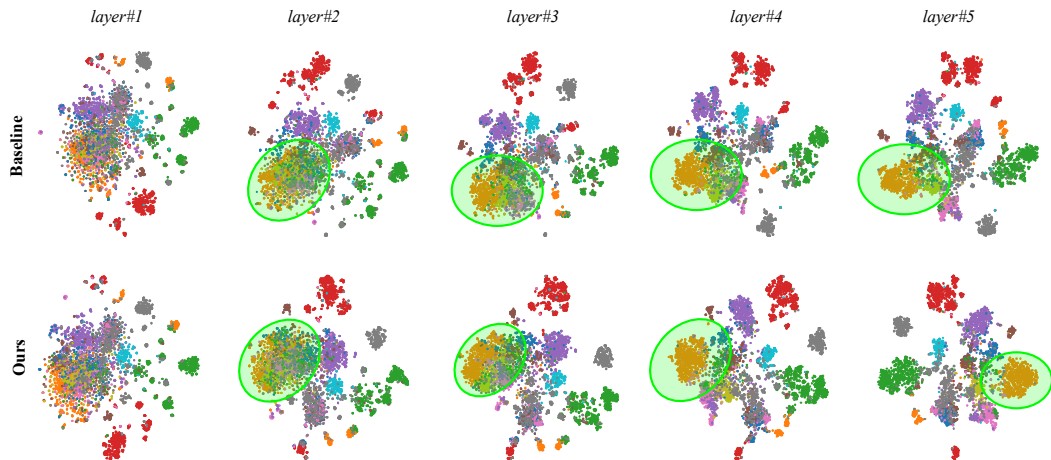

Figure 4: The visualization of graph representation learning on the Roman-empire dataset after t-SNE dimension reduction. Each color represents a distinct node class. A comparison with the baseline (SAGE) reveals that NLE significantly enhances the representation as the model depth increases, particularly in the highlighted region.

### 4.2.4 ABLATION STUDIES

We conduct experiments with various combinations of $K_{start}$ and $K_{end}$, where $K_{start} \in [1, 4]$ and $K_{end} \in [2, 7]$, as detailed in Appendix B. In summary, we found that the combination of $K_{start} = 1$ and $K_{end} = 2$ consistently performed the best across most graph datasets with 5-layer baseline models. (1) Increasing $K_{end}$ while keeping $K_{start} = 1$ led to improved performance on heterophilous graphs whose $k$ steps distant nodes consistently contribute positive non-local information, e.g., Wisconsin ($h$=0.20) and Cornell ($h$=0.13) have rising curves when $K_{end}$ from 1 to 3 in Figure 6. Given that some data can perform better when $k$=4 but not 3, such as Roman-empire and Minesweeper, $K_{start}$=1 and $K_{end}$=2 is the safest setting to obtain better results. Conversely, the impact on homophilous graphs (e.g., Questions ($h$=0.84), Cora ($h$=0.81), Tolokers ($h$=0.60), and Amazon-ratings ($h$=0.38)) was relatively minor, as nodes of the same class tend to be closely connected. (2) Increasing $K_{start}$ from 1 to 4 while maintaining a fixed range has no discernible benefit on Roman-empire ($h$=0.05) when $K_{start}$ increasing accuracies dropped from 83.74% to 75.97% and from 90.34% to 84.81% as listed in Table 6. The NLE with $k \geq 5$ only showed notable improvement on synthetic data Minesweeper, from 86.54% to 90.82% and 87.09 to 92.93%. In light of that, on real-world data, it's not worth applying NLE to very distant nodes since there is no promising improvement of all real-world data as listed in Table 6. (3) Additionally, the learning performance between the cascaded and the aggregated methods, as shown in Table 2, does not manifest significant difference.

## 5 CONCLUSION

In this work, we address the challenge of graph feature representation from a novel perspective of topological re-wiring. We put the spotlight on the efficiency of message-passing mechanism in graph-based deep models to the extent that global information can be effectively captured through a set of express connections between two distant but topologically connected nodes in the graph. By doing so, the re-wired express connection allows us to learn global feature representation while reducing the chance of over-smoothing features in the conventional node-after-node graph diffusion process. Following this notion, we present an *express messenger* wrapper, serving as an agnostic GNN plug-in, to progressively re-wire the graph topology via non-local information exchange, which is reminiscent of non-local mean technique prevailing in image processing area more than a decade ago. In practice, our *ExM* wrapper can substantially enhance the expressibility of graph topology and instantly boost the learning performance for current GNN models. Experiments show that our *ExM* wrapper has achieved SOTA performance on nine graph datasets, indicating the great potential in other graph learning applications such as brain connectomes and drug medicine data.

Our future work includes (1) extensive benchmarks with existing GNN models and public datasets, (2) studying learnable aggregation way of non-local information exchange, and (3) having a deeper insight into the interplay between topological re-wiring, graph expressive power, and graph learning.

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

APPENDIX

## A  DATA STATISTICS

In Table 4, we show the profile of all graph datasets used in this paper.

Table 4: Graph data profiles.

|  | $h$ | Node number | Edge number | Class number |
|---|---|---|---|---|
| Questions | 0.84 | 48,921 | 153,540 | 2 |
| Cora | 0.81 | 2,708 | 10,556 | 7 |
| Pubmed | 0.80 | 19,717 | 88,648 | 3 |
| Citeseer | 0.74 | 3,327 | 9,104 | 6 |
| Minesweeper | 0.68 | 10,000 | 39,402 | 2 |
| Tolokers | 0.60 | 11,758 | 519,000 | 2 |
| Amazon-ratings | 0.38 | 24,492 | 93,050 | 5 |
| Chameleon | 0.24 | 2,277 | 31,421 | 5 |
| Squirrel | 0.22 | 5,201 | 198,493 | 5 |
| Wisconsin | 0.20 | 251 | 515 | 5 |
| Cornell | 0.13 | 183 | 298 | 5 |
| Texas | 0.11 | 183 | 325 | 5 |
| Roman-empire | 0.05 | 22,662 | 32,927 | 18 |

## B  DETAILED ANALYSIS IN ABLATION STUDIES

We show the results of three ablation studies, in addition to experiments in the main text, to support our claims in Section 4.2.4. Figure 5 shows layer number has less influence on performance than model architecture. Please refer to the section 4.2.4 in the main text to see our description for results Figure 6 and Table 6.

Figure 5: Bar plots of performance by using different layer numbers.

Figure 6: Performance with cascaded *ExM* and increasing $K_{end}$, where $K_{start} = 1$ and baselines have 10 layers.

Table 5: Performance with cascaded *ExM* using different $[K_{start}, K_{end}]$, where baselines have 10 layers.

|  | $[K_{start}, K_{end}]$ | Roman. | Amazon. | Mine. | Squirrel | Citeseer |
|---|---|---|---|---|---|---|
| GAT+C-*ExM* | $[1, 4]$ | 83.74±0.55 | 46.44±0.42 | 86.54±0.78 | 61.55±1.33 | 74.25±1.27 |
|  | $[2, 5]$ | 76.44±0.87 | 46.83±0.72 | 90.56±0.59 | 61.48±0.99 | 73.58±1.97 |
|  | $[3, 6]$ | 76.19±0.82 | 46.76±0.71 | 90.96±0.74 | 62.51±0.81 | 73.84±1.27 |
|  | $[4, 7]$ | 75.97±0.70 | 47.00±0.61 | 90.82±0.82 | 61.91±1.32 | 74.25±1.52 |
| SAGE+C-*ExM* | $[1, 4]$ | 90.34±0.42 | 50.25±0.49 | 87.09±0.79 | 43.93±1.21 | 75.30±1.74 |
|  | $[2, 5]$ | 85.00±0.52 | 50.44±0.79 | 92.51±0.40 | 43.30±1.86 | 74.57±1.50 |
|  | $[3, 6]$ | 85.11±0.50 | 51.30±0.52 | 92.81±0.39 | 44.85±1.60 | 74.64±1.72 |
|  | $[4, 7]$ | 84.81±0.50 | 51.59±0.39 | 92.93±0.40 | 44.76±1.79 | 74.35±1.37 |

Table 6: Performance of cascaded vs. aggregated *ExM* on heterophilous graph dataset, where baselines have 3 layers.

|  | Roman. | Amazon. | Chameleon | Squirrel |
|---|---|---|---|---|
| Avg degree | 1.45 | 3.80 | 13.80 | 38.16 |
| GAT+C-*ExM* | 80.77±0.45 | 48.43±0.44 | 70.88±1.60 | 76.00±0.95 |
| GAT+A-*ExM* | 79.99±0.49 | 48.17±0.53 | 73.18±2.08 | 77.52±0.86 |
| FSGNN+C-*ExM* | 72.61±0.57 | 44.56±0.59 | 75.99±1.31 | 66.45±4.17 |
| FSGNN+A-*ExM* | 72.17±0.35 | 45.42±0.45 | 77.76±0.94 | 73.85±2.08 |

