# OpenReview forum: "Efficient Graph Representation Learning by Non-Local Information Exchange"
_ICLR.cc/2024/Conference — ICLR 2024 Conference Withdrawn Submission_

### Official Review · Reviewer_TumZ · 2023-10-31

**Soundness:** 3 good
**Presentation:** 3 good
**Contribution:** 3 good
**Rating:** 5
**Confidence:** 3

**Summary:**

The paper introduces "express messenger" a model-agnostic plug-in that re-wires the learning graph such that new connections between distant nodes are formed. The aim behind this approach is to achieve better non-local information exchange. The authors claim that their module reduces/eliminates the feature over-smoothing issue. They also integrate the plug-in with existing GNNs and demostrate its effectiveness.

**Strengths:**

Figure 1 clearly illustrates the idea discussed in this work and aids in the visualization of the proposed methodology

The experiments that are designed to show the effectiveness of the express messenger are thorough

**Weaknesses:**

(Table 2) Many of the numbers reported for baselines and the proposed method are within the margin of error of each other. This detracts from the effectiveness of the proposed methodology.

**Questions:**

(Table 3) Going by the pattern of improvement in numbers when the ExM plug-in is included; would the authors be comfortable in agreeing to this?: If the delta of performance between GNN and GNN + ExM is plotted against h (x-axis), it would be an elbow curve with the delta approaching zero as h -> 1 and the elbow being somewhere around h = 0.2. Hope this is clear!

---

> ### Author Response · Authors · 2023-11-15
>
> **Weakness:** We have performed two-sample t-test to examine whether the improvement is statistically significant, while compared to other graph rewiring methods DropEdge, GDC, and SDRF are not as significant as ours in Table 2. This can show the effectiveness of our proposed graph rewiring method.
>
> **Question:** Based on datasets we are using, we partially agree with this reviewer regarding the pattern of learning performance w.r.t. h. Since the number of heterophilous graph dataset (with small h ratio) is much less than homo graph (with big h ratio) in graph learning field, Such observation might be biased by the unbalanced datasets.
>
> Another explanation for this pattern could be: Since smaller h ratio leads to far-away nodes bear the same class label, non-local information can promote message aggregations between non-local neighbors to the extent that far-away nodes are instantly connected via the express links. Please check the evidence on our new Fig 2, where the over-smoothness issue has been greatly mitigated by using non-local information.

---

### Official Review · Reviewer_YKr6 · 2023-10-31

**Soundness:** 1 poor
**Presentation:** 1 poor
**Contribution:** 2 fair
**Rating:** 3
**Confidence:** 4

**Summary:**

In this submission, the authors propose a new method for GNN learning that focuses on addressing the challenge of over-smoothing in feature representation. Incorporating a novel mechanism called Non-Local Information Exchange (NLE), enhances the ability of GNNs to combine local and global information effectively. The main contribution is the ExM wrapper, which can be integrated with various GNN models to maintain state-of-the-art performance across different graph datasets, particularly improving performance on heterophilous graphs.

**Strengths:**

1. The paper introduces a new idea to tackle the over-smoothing issue.

2. The experimental results seem interesting. Specifically, the ExM wrapper developed aids most baseline graph neural network (GNN) methods in retaining state-of-the-art (SOTA) performance across various graph datasets with diverse homophily ratios. It is highlighted that the C-ExMP variant of the ExM wrapper often outperforms its counterparts, securing top-3 rankings in multiple datasets​.

**Weaknesses:**

1. The presentation has a large space for improvement. The reviewer does not think the proof makes sense.

2. The theoretical justification of the proposed method is weak. It is unclear why the proposed wrapper can be applied to general models.

3. Some claims in this paper are too strong. For example, the paper mentioned the expressiveness of GNN has not been explored. However, there are many papers focusing this area including the papers cited in this submission.

**Questions:**

Overall, the reviewer thinks the submission is not ready for publication. There are presentation issues and the methodology needs a theoretical justification.

Q1. The proof of Proposition is so unclear, why the matrix operations can be linked with sets(these neighbors)?

There are grammar issues including:

1. State-of-the-arts -> state-of-the-art
2. far-reach neighborhoods -> far-reaching
3. over-smoothing issue -> issues.

---

> ### Author Response · Authors · 2023-11-15
>
> **Weakness 1:** The math and the proposition show why the convergence of our graph re-wiring method is 1-A. It’s for edge modification, not for graph feature computation.
>
> **Weakness 2:** The proposed wrapper is a graph re-wiring method. It is not a part of graph computation but the data augmentation by modifying graph. Thus, it can be plugged into any GNN by just inputting the modified graph data.
>
> **Weakness 3:** We didn’t mention that “the expressiveness of GNN has not been explored”. Our claim is “the expressibility of graph data has not been explored”.
>
> **Question 1:** The proposition shows why hop(A, k) is converging to 1-A. Since we treat A as a binary matrix, we used the set operation for those binary elements. It’s indeed confusing reader that matrix cannot play with set operation. We revised the equation of the proposition in the manuscript by using logic operation: $1+1 = 0$, and $0+1 = 1$. Then
> $$
> 1-A = \mathcal N^K-\mathcal N^{K-1}+…+\mathcal N^1-\mathcal N^0 = \sum hop(A, K) ,	where\ hop(A,k)=(\mathcal N^k-\mathcal N^{k-1}) \in\{0,1\}
> $$
>
> Thank you for pointing out grammar issues, we have revised them.

---

### Official Review · Reviewer_JpoC · 2023-11-01

**Soundness:** 3 good
**Presentation:** 2 fair
**Contribution:** 2 fair
**Rating:** 5
**Confidence:** 4

**Summary:**

This paper proposes a novel framework to enhance the expressive power of GNNs and mitigate the over-smoothing in deep GNNs. The authors introduce an innovative non-local information exchange mechanism inspired by non-local mean techniques from image processing. This mechanism directly connects distant nodes, bypassing the traditional sequential propagation of information. Two express messenger wrapper are proposed to rewire the connection. The two wrapper allows to capture global representations thus free of over-smoothing. Extensive experiments are conducted to validate the effectiveness of the proposed method.

**Strengths:**

1. The commitment to addressing the over-smoothing issue and improve expressiveness in Graph Neural Networks (GNNs) is highly commendable and worthy of research.

2. Extensive experimental validation demonstrates the method's capability in enhancing performance.

**Weaknesses:**

1.	I disagree with the claim made in the abstract that "However, little attention has been paid to improving the expressive power of underlying graph topology." In fact, there has been a significant amount of research in recent years on the expressive power of graph neural networks, which is a topic that this paper lacks discussion on.

2.	In particular, the approach presented in this paper bears similarities to k-hop GNN. Therefore, it is important to provide a detailed discussion and conduct experimental comparisons between the two.

a)	Nikolentzos, Giannis, George Dasoulas, and Michalis Vazirgiannis. "k-hop graph neural networks." Neural Networks 130 (2020): 195-205.

b)	Feng, Jiarui, et al. "How powerful are k-hop message passing graph neural networks." Advances in Neural Information Processing Systems 35 (2022): 4776-4790.

3.	The diagrams included in the article are difficult to comprehend.

4.	The paper lacks experimental or theoretical evidence to support the claim that extracting global structural information can effectively resolve the issue of over-smoothing.

**Questions:**

1.	Why we need to use proxy measurements instead of existing measurement methods？

2.	Could you please explain the design differences and suitable scenarios for the two types of messenger wrapper.?

3.	The ablation study lacks a comparison that involves aggregation of global information exclusively. The overall experiments do not directly demonstrate the impact of different methods on over-smoothing. It is recommended that this be supplemented.

4.	How does capturing global information overcome the problem of over-smoothing? Could an excessive focus on extracting global information potentially lead to even greater over-smoothing?

---

> ### Author Response · Authors · 2023-11-15
>
> **Weakness 1:** As we mentioned in the global reply, we didn’t discussed the expressive power of graph neural networks, but the expressibility of graph data itself. We have replaced “the expressive power of underlying graph topology” with “the potential of a graph to be expressed”.
>
> **Weakness 2:** There are 2 main differences between k-hop GNN and our graph rewiring method:
> (1)	Enhance GNN model (by k-hop) vs enhance the wiring topology (by our method). Our objective is to re-wire the graph to only has edges with k-hop nodes regardless of GNN architecture to learn graph representation more efficient by non-local information, while k-hop GNN is to find a combination for message from 0-hop to k-hop nodes to build a new GNN architecture. For instance, the proposed KP-GNN is a specific GNN framework by building peripheral subgraph.
> (2)	Message passing (by k-hops) vs rewiring (by our method).  Our idea is to progressively modify the graph to connect with k-hop neighbor as the input to the k-th layer of any GNN with any message aggregation, while k-hop GNN is to design a message aggregation for all neighbors from 0-hop to k-hop in each layer of GNN. Although its variant KP-GNN is building subgraph, such modification is more like the graph partition not the graph re-wiring.
> Since models of two papers reviewer mentioned are both designed for graph level representation, we need to adapt their models to node level representation in our experiments at this point and then report the comparison result in the final version.
>
> **Weakness 3:** We have revised the diagram of our method Fig 3 to mainly show cascaded wrapper (C-ExM) and aggregated wrapper (A-ExM), and removed adjacency matrix examples in the top of Fig 3. Please see the revised version of the manuscript.
>
> **Weakness 4:** Oversmoothing issue is present when GNN model becomes deeper and deeper. While deeper GNN can reach far-away nodes, but node features are often excessively smoothed before message passing reaches them. That is the motivation that we propose to capitalize on non-local (or you can call it global) information exchange by creating edges between far nodes and removing old edges, so they can do message passing before the feature is oversmoothed. We have run the same exp as [1, 2] to show Dirichlet energy trends along with layer number increasing on the revised version. Non-local (global) information is helpful to mitigate over-smoothing issue as shown in the new Fig 2 of the revised manuscript.
>
> **Question 1:** The existing measurement is for the expressive power of GNN. There is no measurement for the expressibility of graph data. So, we based on the WL algorithm proposed a proxy measurement for the expressibility.
>
> **Question 2:** Here are two short sentences to describe our wrappers: Cascade ExM wrapper is using non-local edges after the layer using the original edges of graph. The aggregated ExM wrapper uses non-local edges and the original edges in the same layer, and then add them together.
>
> For different scenarios, results as we shown in Table 2 and 3 indicated non-local information exchange is more helpful for heterophilous graphs than homophilous graphs. Thus, we add a new ablation study in supplementary (Table 6) of the revised manuscript, the results of them on different scenarios (different average degree ratio of graph data) indicate the two wrappers are similar for graphs with lower average degree, while A-ExM performs better than C-ExM for graphs with higher average degree. This is a reasonable observation that cascaded and aggregated ExM provide similar information since it makes no big difference of non-local information by cascading with each other layer or aggregating with every layer from few neighbors, and for nodes with higher degree, it is more suitable to use the aggregated wrapper, which rewires graph to non-local neighbors on every layer of GNN to address the big number of neighboring nodes.
>
> **Question 3:** The new Fig 2 has the results of Dirichlet energy under the same exp setup as [1,2]. The trending of oversoomthness is much slower than counterpart methods. While GCN, and GraphSAGE is falling below $10^{-2}$ at 10th layer, they can hold the energy until 100th layer using our ExM. Even for the G2GNN that is not oversmoothing at all, using NLE leads to the energy occasionally increased in layers deeper than 100. For NAGphormer, which is a tokenized graph Transformer model, the energy is also increased using non-local information since the 1st layer.
>
> **Question 4:** We add experimental evidence to support the claim that extracting global structural information can effectively resolve the issue of over-smoothing. Please refer to the reply of Question 3 and  Weakness 4.
>
> [1] Rusch, T. Konstantin, et al. "Gradient gating for deep multi-rate learning on graphs." ICLR 2023.
>
> [2] Rusch, T. Konstantin, et al. "Graph-coupled oscillator networks." International Conference on Machine Learning. PMLR, 2022.

---

### Author Response · Authors · 2023-11-15
**Clarify one important misunderstanding**

We would like to clarify the confusion raised by two reviewers (JpoC & YKr6)) that the proposed term “expressibility” of a graph refers to the potential of a graph to be expressed by any GNN, which is an intrinsic characteristic of the underlying graph data. It is not the expressive power of GNN model, which is a property of model. The reason we proposed this term is that graph rewiring approaches are implemented as a data augmentation that changes the data. While many related works discussed the expressive power of the GNN model, none of metric can quantify how the potential of graph data is affected after graph is modified.

In the manuscript, we claimed the graph rewiring would change such potential, and we first quantified it by our proxy measurement of “graph expressibility” based on WL algorithm. This topic is not discussed by previous graph rewiring works as we cited and discussed in Sec 2. In Fig 2 and Table 1, we also show how expressibility affects the accuracy of node classification on various graph data, where more expressibility normally leads to higher accuracy. And how the expressibility increased by using our re-wiring method, the non-local Exchange Messenger (ExM) wrapper, while previous graph re-wirings, such as DropEdge and GDC, rarely increase the expressibility.
Our findings can be concluded that increasing the expressibility of graph by re-wiring the graph is a new route to reach SOTA results and to mitigate the over-smoothing issue instead of designing a new GNN architecture.

We apologize that we didn’t clearly present the above idea in the Abstract towards to reviewers considering our focus is on the GNN expressive power, while we are not working on the GNN model but the graph data modification.

We revised the manuscript to make our focus clearer on Abstract and Sec 2. We also add two experiments to show some evidence to support (1) non-local information can effectively mitigate the oversmoothing issue, and (2) an additional ablation study on supplementary to demonstrate when to use cascaded (C-ExM) or aggregated (A-ExM) wrapper.